# TASK AND MODEL AGNOSTIC DIFFERENTIALLY PRIVATE GRAPH NEURAL NETWORKS VIA COARSENING

## ABSTRACT

Graph Neural Networks (GNNs) have emerged as powerful tools for analyzing graph-structured data, deriving representations by aggregating information from neighboring nodes. However, this aggregation process inherently increases the risk of exposing confidential data, as a single node may influence the inference process for multiple nodes simultaneously. To mitigate this risk, researchers have explored differentially private training methods for GNN models. Existing privacy-preserving approaches, however, face significant challenges. They often incur high computational costs during training or struggle to generalize across various GNN models and task objectives. To address these limitations, we introduce _Differentially Private Graph Coarsening (DPGC)_, a novel method that tackles two key challenges in GNN training: _scalability_ and _privacy guarantees_ that are independent of the downstream task or GNN model. Through comprehensive experiments on six datasets across diverse prediction tasks, we demonstrate that DPGC sets new benchmarks in graph coarsening. Our method achieves superior compression-accuracy trade-offs while maintaining robust privacy guarantees, outperforming state-of-the-art baselines in this domain.

## 1 INTRODUCTION AND RELATED WORKS

Graph-structured data derived from social and communication networks have become invaluable for generating insights across fields such as social, behavioural, and information sciences Fan et al. (2019); Liu et al. (2024). Such data naturally conforms to a graph-based model, encapsulating rich, nuanced, and organized information. However, there are significant challenges when dealing with large graph data, particularly _scalability_ and _privacy preservation_. Additionally, due to seamless data collection, often facilitated via personal devices, individual data within network contexts tends to be highly sensitive. The strong inter-node relationships in graph-structured data make it especially vulnerable to privacy attacks, increasing the risk of disclosing individuals' data without their consent Liu et al. (2016); Zhang et al. (2021); Mueller et al. (2024).

Beyond privacy concerns, the sheer size of graphs presents significant scalability challenges in graph-based learning methods, as highlighted by recent research Hashemi et al. (2024). Large graphs, comprising millions or even billions of nodes and edges, impose enormous computational and memory demands, making it difficult to perform training and inference with standard methods. The irregular and non-Euclidean nature of graph data exacerbates these challenges, requiring complex operations like neighbourhood aggregation that do not parallelize efficiently. As the field evolves, addressing the scalability of large graphs remains a critical area of research, driving the development of more efficient graph-based learning frameworks.

### 1.1 GAPS IN EXISTING WORKS

**Differential Privacy for GNNs:** Depending on the level of privacy sought, one may consider _edge-level_ privacy, _node-level_ privacy Raskhodnikova & Smith (2016), or both. Edge-level privacy ensures that two graphs differing by a single edge are indistinguishable based on the algorithm's output Blocki et al. (2012); Dwork et al. (2014b); Hardt & Roth (2012); Upadhyay (2013). Alternatively, node-level privacy protects the privacy of individual nodes, a focus of prior studies Daigavane et al. (2021); Olatunji et al. (2023); Zhang et al. (2024). Existing algorithms typically guarantee differential privacy (DP) on the learned GNN embeddings. However, this approach has a significant limitation: the embeddings are often specific to particular GNN architectures and/or loss functions tailored to target tasks. This specificity constrains the flexibility of these algorithms, making it challenging to adapt

Table 1: Comparison of Graph Coarsening Algorithms. The cell in Green indicates the presence of a feature, and the cell in Red indicates its absence. $n$ and $m$ are the number of nodes and edges in the original graph, and $k$ denotes the number of (super) nodes in the coarsened graph.

| Algorithm | Topology-Aware | Feature-Aware | Time Complexity | Privacy |
|---|---|---|---|---|
| LVN Loukas & Vandergheynst (2018) | ✓ | ✗ | $O(n^3)$ | ✗ |
| LVE Loukas & Vandergheynst (2018) | ✓ | ✗ | $O(n^3)$ | ✗ |
| LVC Huang et al. (2021) | ✓ | ✗ | $O(n^3)$ | ✗ |
| HEM Ron et al. (2011) | ✓ | ✗ | $O(m)$ | ✗ |
| Alg. Distance Chen & Safro (2011) | ✓ | ✗ | $O(m)$ | ✗ |
| Affinity Livne & Brandt (2012) | ✓ | ✗ | $O(m)$ | ✗ |
| Kron Dorfler & Bullo (2012) | ✓ | ✗ | $O(n^3)$ | ✗ |
| FGC Kumar et al. (2023) | ✓ | ✓ | $O(n^2 k)$ | ✗ |
| FACH Kataria et al. (2023) | ✓ | ✓ | $O(n)$ | ✗ |
| LAGC Kumar et al. (2024) | ✓ | ✓ | $O(n^2 k)$ | ✗ |
| DPGC (our) | ✓ | ✓ | $O(n)$ | ✓ |

them to novel GNN architectures or unforeseen tasks. In our work, we overcome this limitation by ensuring privacy guarantees irrespective of the GNN architecture or the downstream task. This approach offers a more versatile and robust privacy-preserving framework for graph machine learning.

**Graph Coarsening:** Table 1 lists the various algorithms for graph coarsening. While several algorithms have been proposed, to the best of our knowledge, none of them are *locality-aware*. Specifically, a message-passing GNN Kipf & Welling (2016a); Hamilton et al. (2017); Xu et al. (2018) of $L$-layers would produce similar embeddings for any pairs of nodes with similar $L$-hop neighborhoods Xu et al. (2018). Importantly, this means even distant nodes may produce similar, or even identical, embeddings. Fig. 1 illustrates such an example. Intuitively, the topological distribution of node labels is similar around $v_1$ and $v_{11}$ in Fig 1, making them indistinguishable by any message-passing GNN. Although existing methods consider network connectivity, they typically focus on *positional* information, where short-range, dense connections are criterion for coars-

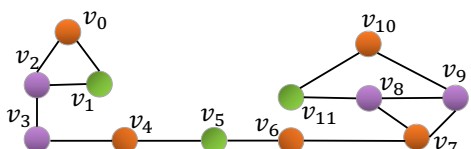

Figure 1: In this shown graph, node labels are represented by colors. Nodes $v_1$ and $v_{11}$ are among the most distant pairs in terms of shortest path length. However, any 2-layer GNN will invariably produce identical embeddings for these nodes. This occurs because their respective 2-hop neighborhoods are indistinguishable under 1-Weisfeiler-Lehman (WL) tests Xu et al. (2018). For a more detailed reasoning, please refer to App. A.1.

ening individual nodes into supernodes. Consequently, locality information, which is the cornerstone of message-passing GNNs, is overlooked. We note that while network proximity often implies similar neighborhoods, the converse may not hold true. Finally, as shown in Table 1, we note that several algorithms also ignore node attribute information during the coarsening process.

## 1.2 CONTRIBUTIONS

In this paper, we propose _Differentially Private Graph Coarsening (DPGC)_, the first unified framework to jointly perform graph coarsening while also ensuring differential privacy for *any* downstream message-passing GNN architecture and tasks. The key innovations driving this result are as follows:

- **GNN-aligned graph coarsening:** Instead of sole reliance on spectral similarity, DPGC coarsens the topology by embedding nodes into a feature space using Weisfeiler-Lehman kernel. Subsequently, the nodes are coarsened into supernodes through locality sensitive hashing. This design aligns DPGC to the computation framework of message-passing GNNs and unshackles itself from the limitations of existing coarsening strategies where distant nodes, even when producing similar embeddings, are not considered as candidates to be merged into a supernode.
- **Strong theoretical guarantees:** DPGC is grounded on rigorous theoretical guarantees. First, we prove that DPGC is $(\varepsilon, \delta)$-*DP* (differentially private). Second, by virtue of *post-processing* theorem, any GNN trained on the coarsened graph produced by DPGC is also differentially private. Third, DPGC ensures *restricted spectral similarity*. Finally, DPGC *scales linearly* with the number of nodes in a graph resulting in superior efficiency (Table 1).
- **Empirical evaluation:** Through extensive experiments on six real-world datasets, we establish that DPGC outperforms state-of-the-art graph coarsening algorithms on accuracy, while ensuring superior privacy-accuracy trade-off compared to existing algorithms for differential privacy on GNNs.

## 2 BACKGROUND AND PROBLEM FORMULATION

In this section, we formulate our problem and introduce the preliminary concepts central to our work.

### 2.1 GRAPH COARSENING

A graph with node features is denoted by $\mathcal{G} = (\mathcal{V}, \mathcal{E}, \mathbf{A}, \mathbf{X})$, where $\mathcal{V} = \{v_1, v_2, \cdots, v_n\}$ is the vertex set, $\mathcal{E} \subseteq \mathcal{V} \times \mathcal{V}$ is the edge set and $\mathbf{A}$ is the adjacency (weight) matrix corresponding to the graph. Let $\mathbf{X} = [\mathbf{x}_1, \cdots, \mathbf{x}_n]^T$, where $\mathbf{x}_i$ is the $d$-dimensional feature vector associated with $i$-th node of an undirected graph. Given an original graph $\mathcal{G} = (\mathcal{V}, \mathcal{E}, \mathbf{A}, \mathbf{X})$ with $n$ nodes, the goal of graph coarsening is to construct an appropriate "smaller" or coarsened graph $\tilde{\mathcal{G}} = (\tilde{\mathcal{V}}, \tilde{\mathcal{E}}, \tilde{\mathbf{A}}, \tilde{\mathbf{X}})$ with $k \ll n$ nodes, such that $\tilde{\mathcal{G}}$ and $\mathcal{G}$ have similar properties. In coarsening, we define a linear mapping $\pi : \mathcal{V} \to \tilde{\mathcal{V}}$ that maps a set of nodes in $\mathcal{G}$ having similar properties to a *super-node* in $\tilde{\mathcal{G}}$, i.e, $\{\pi^{-1}(\tilde{v}) : \tilde{v} \in \tilde{\mathcal{V}}\}$ is a partition of $\mathcal{V}$.

Let the Laplacian matrices of the graphs $\mathcal{G}$ and $\tilde{\mathcal{G}}$ be denoted as $\mathbf{L} \in \mathbb{R}^{n \times n}$ and $\tilde{\mathbf{L}} \in \mathbb{R}^{k \times k}$, respectively. Following Loukas (2019) we define the *coarsening matrix* $\mathbf{P} \in \mathbb{R}_+^{k \times n}$ and call its pseudo-inverse $\mathbf{C} = \mathbf{P}^\dagger$ as the *loading matrix*. These matrices are *Laplacian-consistent*, i.e., they are defined such that:

$$\tilde{\mathbf{L}} = \mathbf{C}^T \mathbf{L} \mathbf{C}.$$

The details on how such a matrix can be chosen are available in Loukas (2019). We use $\tilde{\mathbf{X}}$ to denote the feature matrix of the coarsened graph.

**Definition 1** (Graph Coarsening for GNNs). *Let $\mathcal{G} = (\mathcal{V}, \mathcal{E}, \mathbf{A}, \mathbf{X})$ be a graph with an associated node feature matrix $\mathbf{X} \in \mathbb{R}^{n \times d}$ and node labels $\mathbf{Y} \in \{0,1\}^{n \times c}$, where $c$ is the number of classes. The goal is to obtain a coarsened graph $\tilde{\mathcal{G}} = (\tilde{\mathcal{V}}, \tilde{\mathcal{E}}, \tilde{\mathbf{A}}, \tilde{\mathbf{X}})$ such that a GNN model $\mathcal{M}(\tilde{A}, \tilde{\mathbf{X}}; \Theta)$, parameterized by $\Theta$ and trained on $\tilde{\mathcal{G}}$, satisfies the following:*

$$\min \mathcal{L} [\mathcal{M}(\mathbf{A}, \mathbf{X}; \Theta), \mathbf{Y}]$$
$$s.t. \quad \Theta = \arg\min_\theta \mathcal{L} [\mathcal{M}(\tilde{\mathbf{A}}, \tilde{\mathbf{X}}; \theta), \tilde{\mathbf{Y}}], \tag{1}$$

*where $\mathcal{L}$ is a loss function, and $\tilde{\mathbf{Y}}$ is the corresponding label set for the coarsened graph obtained as the majority label of constituent nodes.*

### 2.2 DIFFERENTIAL PRIVACY

Differential Privacy (DP) ensures that the output distributions of an algorithm remain indistinguishable, with a specific probability, when the input datasets differ by only a single record. The datasets that differ by one record are called *neighbours*, and in the case of graph data, the neighbour graph can be defined in teams of node and edge difference. For edge-privacy, we call two graphs $\mathcal{G} = (\mathcal{V}, \mathcal{E})$ and $\mathcal{G}' = (\mathcal{V}', \mathcal{E}')$ neighbouring if it holds that $\mathcal{V} = \mathcal{V}'$ and $|(\mathcal{E} \backslash \mathcal{E}') \cup (\mathcal{E}' \backslash \mathcal{E})| \leq 1$. For node-privacy, the graphs $\mathcal{G} = (\mathcal{V}, \mathcal{E})$ and $\mathcal{G}' = (\mathcal{V}', \mathcal{E}')$ are neighbouring if they differ by a single node and its corresponding edges. Then, the definition of edge or node differential privacy can be given as follows:

**Definition 2** (Differential Privacy Dwork et al. (2014a)). *A randomized algorithm $\mathcal{A}$ is $(\varepsilon, \delta)$-differentially private if for all neighboring graphs $\mathcal{G}$ and $\mathcal{G}'$ and all subsets of outputs $S$,*

$$\Pr[\mathcal{A}(\mathcal{G}) \in S] \leq e^\varepsilon \cdot \Pr[\mathcal{A}(\mathcal{G}') \in S] + \delta,$$

*where the probability is over the randomness of the algorithm.*

Here, $\varepsilon$ is the *privacy budget*: a lower privacy budget leads to stronger privacy guarantees but reduced utility. $\delta$ is the *failure probability* and is usually chosen to be very small.

**Privacy Attacks on GNNs: Membership Inference Attacks.** In a membership inference (MI) attack, an adversary tries to infer whether a data point was part of the training set used to train the target model. An MI attack on GNNs can be defined as follows:

**Definition 3** (MI Attack on GNN). *Let a GNN model $\mathcal{M}$ be trained using the graph $\mathcal{G}_t = (\mathcal{V}_t, \mathcal{E}_t)$. Given a node $v$ and its $L$-hop neighborhood, the adversary aims to determine whether $v \in \mathcal{V}_t$. Note that even if $v$ was part of the training set, the $L$-hop neighborhood known to the adversary might differ from the one used during the training of the model $\mathcal{M}$.*

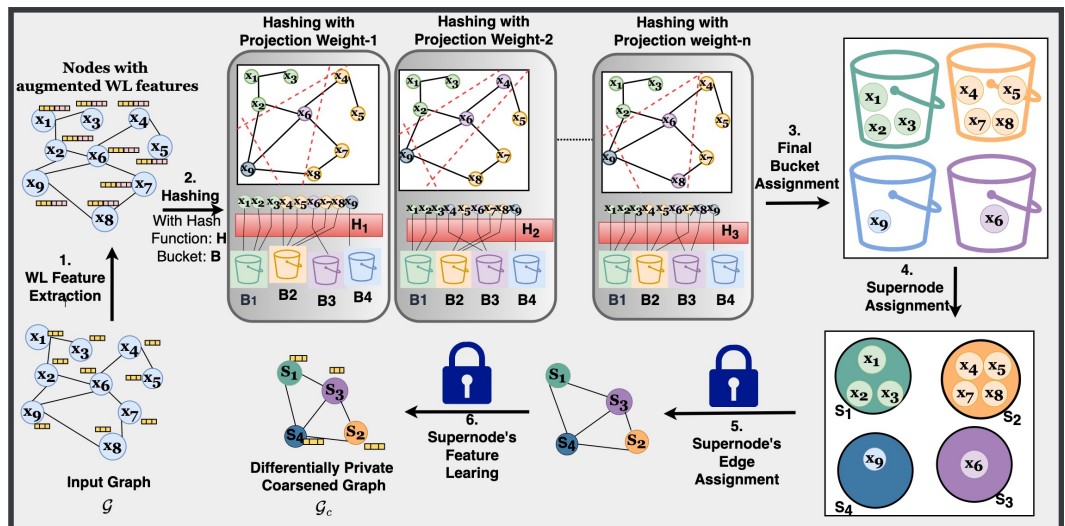

Figure 2: The figure illustrates the overall pipeline of the DPGCframework.

In other words, the adversary constructs a classifier $\mathcal{C}(f_\theta(v))$ for node $v$, such that:

$$\Pr[\mathcal{C}(f_\theta(v)) = 1 \mid v \in \mathcal{V}_t] > \Pr[\mathcal{C}(f_\theta(v)) = 1 \mid v \notin \mathcal{V}_t].$$

Similarly, an adversary can design an MI attack for an edge $e \in \mathcal{E}_t$. To mitigate these MI attacks, designing differentially private graph neural networks (DP-GNNs) is a de facto solution. The problem of building DP-GNNs can be defined as follows:

**Definition 4** (DP-GNNs). *Let $\mathcal{G} = (\mathcal{V}, \mathcal{E}, \mathbf{A}, \mathbf{X})$ represent a graph, where $\mathbf{X} \in \mathbb{R}^{n \times d}$ is the node feature matrix, and $\mathbf{A}$ is the adjacency matrix. The goal is to train a GNN model $\mathcal{M}(\mathbf{A}, \mathbf{X}; \Theta)$ such that it satisfies differential privacy (DP), ensuring that:*

1. *The model's outputs ($o$) on any two neighboring graphs $\mathcal{G}$ and $\mathcal{G}'$, differing by a single node or edge, are indistinguishably close with respect to a privacy parameter $\varepsilon$.*
2. *The model $\mathcal{M}$ can still perform effectively on downstream tasks, such as node classification or link prediction, while providing rigorous privacy guarantees.*

*Mathematically, the model must satisfy the following DP condition:*

$$Pr[\mathcal{M}(\mathcal{G}) = o] \leq e^\varepsilon \cdot Pr[\mathcal{M}(\mathcal{G}') = o] + \delta, \tag{2}$$

*where $\varepsilon > 0$ is the privacy budget, and $\delta$ is a small failure probability.*

### 2.3 PROBLEM FORMULATION

**Problem 1** (**Differentially Private Graph Coarsening**). *Let $\mathcal{G} = (\mathcal{V}, \mathcal{E}, \mathbf{A}, \mathbf{X})$ be a graph with an associated node feature matrix $\mathbf{X} \in \mathbb{R}^{n \times d}$ and node labels $\mathbf{Y} \in \{0,1\}^{n \times c}$, where c is the number of classes. Our goal is to construct a coarsened graph $\tilde{\mathcal{G}}$ that satisfies the quality objectives of Def. 1, while being differential private as defined in Def. 3.*

## 3 DPGC: PROPOSED METHODOLOGY

Fig. 2 presents the pipeline of the proposed algorithm DPGC. There are three key steps:

1. **Unsupervised node embeddings:** Each node is embedded into a feature space using the *Weisfeiler Lehman Kernel* (Togninalli et al., 2019).
2. **Supernode construction:** Next, these embeddings are hashed into buckets using *Locality Sensitive Hashing (LSH)* (Charikar, 2002). Based on the collisions obtained in LSH, the nodes are grouped into supernodes, and then edges and are determined among these supernodes.
3. **Learning supernode attributes:** Finally, the attributes of the supernodes are generated to obtain the differentially private coarsened graph.

In the subsequent sections, we detail each of the individual steps and establish how differential privacy is ensured in DPGC.

## 3.1 Augmented Weisfeiler-Lehman Node Features

Intuitively, we seek to group nodes into a supernode if their embeddings are similar. GNNs follow an iterative message-passing scheme where the embedding of a node is a function of its $L$-hop ego graph; $L$ represents the number of layers in the GNN. Formally, the node embeddings are computed as follows:

**Initialization:** Set $\mathbf{h}_i^0 = \mathbf{x}_i, \forall v_i \in \mathcal{V}$.

**Message creation:** In layer $\ell$, *collect* and *aggregate* messages from each neighbor.

$$\mathbf{m}_i^\ell(j) = \text{Message}^\ell(\mathbf{h}_j^{\ell-1}, \mathbf{h}_i^{\ell-1}), \quad \forall v_j \in \mathcal{N}_i = \{v_j \mid (v_j, v_i) \in \mathcal{E}\}$$

$$\overline{\mathbf{m}}_i^\ell = \text{Aggregate}^\ell(\{\{\mathbf{m}_i^\ell(j) : v_j \in \mathcal{N}_i\}\})$$

**Update embedding:** $\mathbf{h}_i^\ell = \text{Update}^\ell(\mathbf{h}_i^{\ell-1}, \overline{\mathbf{m}}_i^\ell)$

$\text{Message}^\ell$, $\text{Aggregate}^\ell$, and $\text{Update}^\ell$ may be predefined functions (e.g., mean pooling) or neural networks. $\{\{\cdot\}\}$ denotes a multiset since different neighbors may send an identical message. This process repeats for $L$ layers to generate the final node representations $\mathbf{z}_i = \mathbf{h}_i^L$.

We now point to two properties of GNNs that have been established in the literature (Xu et al., 2018).

**Definition 5** (Sufficiency)**.** *In an $L$-layered GNN, the $L$-hop ego graph is sufficient to compute node embedding $\mathbf{z}_i = \mathbf{h}_i^L$, $\forall v_i \in \mathcal{V}$.*

**Definition 6** (Equivalence)**.** *The expressive power of a message-passing GNN is limited by the Weisfeiler-Lehman (1-WL) test (Xu et al., 2018). This implies that if the $L$-hop ego graphs of two nodes cannot be distinguished by the 1-WL test, then their corresponding embeddings will be identical. Importantly, the 1-WL test fails to differentiate between isomorphic graphs (Shervashidze et al., 2011).*

Empowered by these observations, we propose mapping nodes with similar $L$-hop ego graphs to a single supernode. To achieve this, we embed each node into a feature vector using the Weisfeiler-Lehman (WL) kernel, a generalization of 1-WL test where similar (ego) graphs generate similar embeddings.

**Definition 7** (Weisfeiler-Lehman (WL) Kernel (Togninalli et al., 2019))**.** *WL-kernel embeds each node in a graph via a message-passing aggregation. Like in GNNs, the initial embedding $\mathbf{a}_i^0$ in layer 0 is $\mathbf{x}_i \in \mathbf{X}$. The embedding in any subsequent layer $\ell$ is defined as:*

$$\mathbf{a}^\ell(i) = \frac{1}{2}\left(\mathbf{a}_i^{\ell-1} + \frac{1}{\deg(v_i)}\sum_{j \in \mathcal{N}(i)} w((i,j)) \cdot \mathbf{a}_j^{\ell-1}\right) \tag{3}$$

$\mathcal{N}(i)$ denotes the neighbors of $v_i$ and $\deg(v_i) = |\mathcal{N}(i)|$. For unweighted graphs, $w((i,j)) = 1$. We denote the final embedding of node $v_i$ after $L$ hops as $\mathbf{a}_i$. We used a 1-hop ego graph, and the final embeddings were obtained by concatenating the original node features to it.

## 3.2 Constructing Differentially Private Supernodes via LSH

To map nodes to supernodes while achieving differential privacy, we use the method of Kenthapadi et al. (2013), i.e., we use *Locality Sensitive Hashing (LSH)* to hash the features associated with each node, adding a bias in order to ensure differential privacy. All nodes whose final value lands in the same bucket are mapped to the same supernode. The specific hash function used is:

**Definition 8** (Random Hyperplane based Hash Function (Charikar, 2002))**.** *Let $\mathbf{x}$ and $\mathbf{y}$ be vectors in $\mathbb{R}^d$, and let $\mathbf{w} \in \mathbb{R}^d$ be a random vector drawn from a $d$-dimensional Gaussian distribution. The hash function $h_\mathbf{w}$ corresponding to $\mathbf{w}$ is defined as:*

$$h_\mathbf{w}(\mathbf{x}) = \begin{cases} 1 & \text{if } \mathbf{w} \cdot \mathbf{x} \geq 0 \\ 0 & \text{if } \mathbf{w} \cdot \mathbf{x} < 0 \end{cases}$$

We note that one of the properties of this hash function is that it tends to hash together those data points that are close to each other in Euclidean space. To apply LSH to graph nodes, we project each node $v_i$'s WL embedding $\mathbf{a}_i$ onto $J$ random vectors, $\mathbf{H} = \{\mathbf{w}_1, \cdots, \mathbf{w}_J\}$, creating $J$ hash tables. Each projection ($\mathbf{w}_j \cdot \mathbf{a}_i$, $\mathbf{w}_j \in \mathbf{H}$) maps $\mathbf{a}_i$ to a real line as a function of their similarity. We divide the projection line into equal-width bins of width $r$ and assign each node to a bin using the scalar projection:

$$h_j(\mathbf{a_i}) = \left\lfloor \frac{\mathbf{w}_j \cdot \mathbf{a_i} + b_i}{r} \right\rfloor$$

where $b_i$ is a random bias drawn from a predefined distribution (e.g., Gaussian or uniform).

In effect, let $\mathbf{H} \in \mathbb{R}^{d \times J}$ be the matrix containing projection vectors, and let $\mathbf{B} \in \mathbb{R}^{n \times J}$ be the matrix of bias terms corresponding to each of the $J$ hash functions, drawn from a Gaussian distribution $b_{ij} \sim \mathcal{N}[0, \sigma^2]$ ($i = 1, 2, \ldots, n$ and $j = 1, 2, \ldots, J$). We will discuss the choice of $\sigma$ below. The hashcode is then obtained as $\mathbf{D} = (\mathbf{X} \cdot \mathbf{H} + \mathbf{B})/r$. It is important to note that the dimension $d$ should be consistent with the dimension of the augmented feature vector. A larger $J$ increases the probability of finding true nearest neighbors, reducing false negatives. On the other hand, a larger $r$ implies wider bins, which may increase false positives. The final hashcode $h_i$ assigned to node $v_i$ is the most frequent hash generated across all $J$ projections.

$$h_i = \arg\max_x |j : D_{ij} = x, 1 \leq j \leq J|$$

We create a supernode corresponding to each unique final hashcode. Subsequently, each node $v_i$ is mapped to the supernode that corresponds to its assigned hashcode $h_i$. Further, we assign an edge between two supernodes, say $\tilde{u}$ and $\tilde{v}$, if any nodes $u$ and $v$, respectively mapped to $\tilde{u}$ and $\tilde{v}$, have an edge in the original graph. This method of creating edges ensures that the coarsening matrix $\mathbf{P}$ satisfies the characterisation of Laplacian consistency given in Proposition 7 of (Loukas, 2019).

We now discuss how to set the standard deviation $\sigma$ of the bias matrix $\mathbf{B}$. It is shown in (Kenthapadi et al., 2013) (Theorem 1) that to achieve $(\epsilon, \delta)$-differentially privacy with $\delta < 1/2$ we need

$$\sigma \geq w_2(\mathbf{H}) \frac{\sqrt{2 \ln(\frac{1}{2\delta}) + \epsilon}}{\epsilon}$$

where $w_2(\mathbf{H})$ is the maximum of the $\ell_2$-norms of the rows of $\mathbf{H}$ (referred to as the $\ell_2$-sensitivity of $\mathbf{H}$ in Kenthapadi et al. (2013).)

## 3.3 Learning Attributes of Supernodes with Similarity Guarantee

Our next objective is to obtain a suitable transformation $F$ such that $\tilde{\mathbf{X}} = F(\mathbf{X})$, i.e., that produces the coarsened graph's feature matrix from the original graph's features. Clearly, $\tilde{\mathbf{X}}$ and $\mathbf{X}$ must be "similar" in some way. We make the notion of similarity concrete by using the concept of *Spectral Similarity for Coarsened Graph Data* (SSCGD) (Kumar et al., 2023). In the following, given a $d \times k$ matrix $\mathbf{X}$ and a $k \times k$ positive definite matrix $\mathbf{A}$, $\|\mathbf{X}\|_{\mathbf{A}} := \mathrm{Tr}(\mathbf{X}^T \mathbf{A} \mathbf{X})$.

**Definition 9** ($\eta$-Spectral Similarity for Coarsened Graph Data). *(Kumar et al., 2023) If $\mathcal{G} = (\mathcal{V}, \mathcal{E}, \mathbf{A}, \mathbf{X})$ is a graph with an associated node feature matrix $\mathbf{X}$ and Laplacian $\mathbf{L}$, and $\tilde{\mathcal{G}} = (\tilde{\mathcal{V}}, \tilde{\mathcal{E}}, \tilde{\mathbf{A}}, \tilde{\mathbf{X}})$ is a coarsened version of this graph with Laplacian $\tilde{\mathbf{L}}$ and associated features $\tilde{\mathbf{X}} = F(\mathbf{X})$ where $F$ is a (not necessarily linear) transformation between the two feature spaces, we say that the coarsening $\tilde{\mathcal{G}}$ is $\eta$-SSCGD to $\mathcal{G}$*

$$(1 - \eta)\|\mathbf{X}\|_{\mathbf{L}} \leq \|\tilde{\mathbf{X}}\|_{\tilde{\mathbf{L}}} \leq (1 + \eta)\|\mathbf{X}\|_{\mathbf{L}}, \tag{4}$$

*for some $\eta \geq 0$.*

A small value of $\eta$ indicates strong preservation of multiple properties between the original graph and its coarsened counterpart. Specifically, the *Courant-Fisher* theorem (Loukas, 2019) implies that the spectrum of the coarsened graph approximates the first $k$ eigenvalues of the original graph. This spectral preservation encodes crucial information about graph cuts and random walk dynamics. Intuitively, the similarity in the values of quadratic forms suggests that the interplay between node features and graph topology, as encoded in the respective Laplacians, is maintained through the coarsening process. This feature-topology interaction preservation is fundamental to the performance of GNNs. The fact that using an $\eta$-SSCGD transformation with $\eta$ bounded in $[0, 1)$ gives us results that match the original GNN validate this intuition. We use the method of Kumar et al. (2023) to learn a feature transformation for which it is possible to bound $\eta$. This is done by minimizing the objective function:

$$\min_{\tilde{\mathbf{X}}} \{ \mathrm{Tr}(\tilde{\mathbf{X}} \mathbf{C}^T \mathbf{L} \mathbf{C} \tilde{\mathbf{X}}) + \frac{\alpha}{2} \|\mathbf{C}\tilde{\mathbf{X}} - \mathbf{X}\|_F^2 \}. \tag{5}$$

Since the Eq. 5 is strongly convex, a closed-form solution exists. However, to avoid the inversion of the matrix, we use a gradient descent updates to find $\tilde{\mathbf{X}}$.

$$\tilde{\mathbf{X}}^{t+1} = \tilde{\mathbf{X}}^t - \xi \left[ 2\mathbf{C}^T \mathbf{L} \mathbf{C} \tilde{\mathbf{X}} + \alpha \mathbf{C}^T (\mathbf{C}\tilde{\mathbf{X}} - \mathbf{X}) \right] \tag{6}$$

$\xi$ is the learning rate, and $\alpha$ is a hyperparameter. We initialize $\tilde{\mathbf{X}}$ as $\tilde{\mathbf{X}} = \mathbf{P}\mathbf{X}$. It is shown in Kumar et al. (2023) that this method gives an $\eta \in [0, 1)$.

## 3.4 CHARACTERIZATION OF DPGC

**Node-DP.** To ensure the privacy of node features in the coarsened graph, we use the output perturbation-based DP framework proposed in (Chaudhuri et al., 2011) (Dwork et al., 2006). In this framework, the non-private outcome is perturbed by adding noise according to the sensitivity of the optimization objective. The $\ell_2$-sensitivity of Eq. 5) is proportional to $\alpha$. Then using Gaussian mechanism, $(\varepsilon, \delta)$-DP node features are given as follows:

$$\tilde{\mathbf{X}}' = \tilde{\mathbf{X}} + \mathcal{N}(0, \sigma^2 \mathbf{I}),$$

where $\sigma \geq k\alpha \frac{\sqrt{2\log(\frac{1}{\delta})}}{\varepsilon}$ and $k$ is the number of supernodes in the coarsened graph.

**Theorem 1** (Differentially private GNN). *The GNN trained on the coarsened graph is DP.*

*Proof.* If we have a differentially private output, any computation performed on the output of this mechanism will not degrade the privacy guarantee. This property is known as *Post-Processing Theorem* (Dwork et al., 2014a). Formally,

**Lemma 1** (Post-Processing Theorem (Dwork et al., 2014a)). *Let $\mathcal{M} : D \rightarrow R$ be an $(\varepsilon, \delta)$-differentially private mechanism, and let $f : R \rightarrow R'$ be any (possibly randomized) function. Then the mechanism $f \circ M : D \rightarrow R'$, defined by $(f \circ M)(D) = f(M(D))$, is also $(\varepsilon, \delta)$- differentially private.*

Since our coarsening is DP, the GNN trained on it is also DP. □

**Generalizability of DP:** Unlike existing differentially private GNNs that offer task-specific or architecture-specific privacy guarantees, DPGC provides a general-purpose privacy guarantee applicable to any GNN architecture and downstream task. This is achieved by ensuring that the coarsening process itself is differentially private, thus guaranteeing the privacy of any subsequent operations on it.
**Computation Cost:** The time complexity of DPGC is $O(nJd + m)$. App. A.2 details the derivation.

## 4 EXPERIMENTS

In this section, we benchmark DPGC and establish:

- **Differential Privacy (DP) Vs. Accuracy trade-off:** DPGC outperforms state-of-the-art algorithms for DP in GNNs by demonstrating higher accuracy across a multitude of prediction tasks, datasets and privacy budgets.
- **Resilience:** DPGC exhibits robust resilience to membership inference attacks (Def. 3).
- **Coarsening vs. Accuracy trade-off:** GNNs trained on coarsened graphs produced by DPGC leads to superior accuracy when compared to existing baselines.

The details of our experimental setup and parameters are detailed in App. A.3.1. Our codebase is available at https://anonymous.4open.science/r/DPGC-6BE8.

### 4.1 DATASETS, BASELINES AND METRICS

**Datasets.** Table 2 characterizes the 6 real-world datasets used to benchmark DPGC.

**Baselines for graph coarsening:** We include 8 state-of-the-art baselines: **Local Variation Edges (LVE)** and **Local Variation Neighbourhood (LVN)** are two variants proposed in Loukas & Vandergheynst (2018); **Local Variation Clique (LVC)** (Huang et al., 2021); **Algebraic Distance**, based on the strength of node connections (Chen & Safro, 2011); **Affinity** (Livne & Brandt, 2012) and **Heavy Edge Matching**

Table 2: Summary statistics of used datasets.

| Name | # nodes | # edges | # Features | # Classes |
|------|---------|---------|-----------|-----------|
| Cora | 2,708 | 5,278 | 1,433 | 7 |
| CiteSeer | 3,327 | 9,104 | 3,703 | 6 |
| PubMed | 19,717 | 44,324 | 500 | 3 |
| Coauthor | 18,333 | 163,788 | 6,805 | 15 |
| Physics | 34,493 | 247,962 | 8,415 | 5 |
| DBLP | 17,716 | 105,734 | 1,639 | 4 |

**(HEM)** (Ron et al., 2011), both of which aggregate nodes based on a proximity measure; **Kron**, which reduces the graph based on the Schur complement of the original Laplacian (Dorfler & Bullo, 2012); and **FACH**, which uses hashing (Kataria et al., 2023).

**Private GNN baselines:** We consider 5 baselines for node-DP and 4 for edge-DP. For node-DP, we include **DP-MLP** (Daigavane et al., 2021), **DP-GNN** (Daigavane et al., 2021), **GAP** (Sajadmanesh et al., 2023), **PrivGNN** (Olatunji et al., 2023) and **DPAR** (Zhang et al., 2024).

For edge-DP, we compare our algorithm with **GAP** (Sajadmanesh et al., 2023), **DPGCN** (Wu et al., 2022), **LPGNet** (Kolluri et al., 2022) and **Eclipse** (Tang et al., 2024).

Table 3: (Node-DP) Test accuracy of DPGC and the baseline method for different privacy budgets ($\varepsilon$) on various graph datasets. The value of $\delta$ is $2 \times 10^{-3}$ for Cora, CiteSeer & PubMed datasets and $1 \times 10^{-4}$ for other datasets. Boldface indicates the best results.

| Dataset | $\varepsilon$ | DP-MLP | DP-GNN | GAP | PrivGNN | DPAR | DPGC |
|---|---|---|---|---|---|---|---|
| Cora | 1 | $27.33 \pm 1.17$ | $29.60 \pm 1.04$ | $34.50 \pm 0.24$ | $36.35 \pm 0.26$ | $34.21 \pm 0.15$ | $\mathbf{67.05 \pm 0.29}$ |
|  | 8 | $41.07 \pm 0.77$ | $45.06 \pm 1.39$ | $57.33 \pm 0.16$ | $56.82 \pm 0.19$ | $61.99 \pm 0.28$ | $\mathbf{73.55 \pm 0.22}$ |
| CiteSeer | 1 | $28.12 \pm 0.25$ | $32.42 \pm 0.42$ | $34.00 \pm 0.46$ | $35.23 \pm 0.15$ | $33.55 \pm 0.81$ | $\mathbf{66.95 \pm 0.25}$ |
|  | 8 | $31.72 \pm 0.47$ | $41.25 \pm 0.23$ | $56.55 \pm 0.42$ | $55.10 \pm 0.22$ | $62.00 \pm 0.20$ | $\mathbf{72.48 \pm 0.36}$ |
| PubMed | 1 | $63.70 \pm 0.60$ | $65.50 \pm 0.45$ | $60.66 \pm 0.26$ | $63.50 \pm 0.40$ | $75.60 \pm 0.28$ | $\mathbf{83.50 \pm 0.22}$ |
|  | 8 | $66.20 \pm 1.08$ | $66.25 \pm 0.45$ | $73.14 \pm 0.14$ | $73.86 \pm 0.14$ | $80.65 \pm 0.22$ | $\mathbf{85.00 \pm 0.10}$ |
| Coauthor | 1 | $51.44 \pm 0.16$ | $55.62 \pm 0.12$ | $66.00 \pm 0.10$ | $67.11 \pm 0.22$ | $89.27 \pm 0.05$ | $\mathbf{91.98 \pm 0.10}$ |
|  | 8 | $63.50 \pm 1.05$ | $63.20 \pm 0.66$ | $85.37 \pm 0.13$ | $84.37 \pm 0.22$ | $90.63 \pm 0.37$ | $\mathbf{93.07 \pm 0.28}$ |
| Physics | 1 | $54.12 \pm 0.81$ | $55.88 \pm 0.60$ | $81.92 \pm 0.59$ | $80.10 \pm 0.35$ | $89.48 \pm 0.24$ | $\mathbf{92.70 \pm 0.15}$ |
|  | 8 | $60.17 \pm 0.75$ | $67.22 \pm 0.66$ | $90.88 \pm 0.24$ | $88.36 \pm 0.28$ | $91.01 \pm 0.35$ | $\mathbf{94.40 \pm 0.10}$ |

Table 4: (Edge-DP) Test accuracy of DPGC and the baseline method for different privacy budgets ($\varepsilon$) on various graph datasets. The value of $\delta$ is $2 \times 10^{-3}$ for Cora, CiteSeer & PubMed datasets and $1 \times 10^{-4}$ for other datasets. Boldface indicates the best results.

| Dataset | $\varepsilon$ | DP-MLP | GAP | DPGCN | LPGNet | Eclipse | DPGC |
|---|---|---|---|---|---|---|---|
| Cora | 1 | $27.33 \pm 1.17$ | $34.50 \pm 0.24$ | $35.80 \pm 0.42$ | $48.05 \pm 0.22$ | $65.70 \pm 0.31$ | $\mathbf{71.66 \pm 0.16}$ |
|  | 8 | $41.07 \pm 0.77$ | $57.33 \pm 0.16$ | $73.10 \pm 1.05$ | $70.60 \pm 0.97$ | $67.00 \pm 0.66$ | $\mathbf{73.80 \pm 0.35}$ |
| CiteSeer | 1 | $28.12 \pm 0.25$ | $34.00 \pm 0.46$ | $34.80 \pm 1.10$ | $49.15 \pm 0.88$ | $63.20 \pm 0.046$ | $\mathbf{67.55 \pm 0.12}$ |
|  | 8 | $31.72 \pm 0.47$ | $56.55 \pm 0.42$ | $63.30 \pm 0.66$ | $64.30 \pm 0.52$ | $63.50 \pm 0.46$ | $\mathbf{73.05 \pm 0.37}$ |
| PubMed | 1 | $63.70 \pm 0.60$ | $60.66 \pm 0.26$ | $54.20 \pm 0.62$ | $67.66 \pm 0.86$ | $73.22 \pm 0.20$ | $\mathbf{83.55 \pm 0.28}$ |
|  | 8 | $66.20 \pm 1.08$ | $73.14 \pm 0.14$ | $63.46 \pm 1.02$ | $75.35 \pm 0.44$ | $72.20 \pm 0.88$ | $\mathbf{85.00 \pm 0.30}$ |
| Physics | 1 | $54.12 \pm 0.81$ | $81.92 \pm 0.59$ | $64.20 \pm 0.32$ | $69.00 \pm 0.66$ | $89.10 \pm 0.12$ | $\mathbf{92.85 \pm 0.10}$ |
|  | 8 | $60.17 \pm 0.75$ | $90.88 \pm 0.24$ | $89.46 \pm 0.16$ | $91.06 \pm 0.08$ | $90.15 \pm 0.11$ | $\mathbf{94.45 \pm 0.05}$ |

**Metrics:** In addition to *accuracy* on test datasets, we use *Relative Eigen Error (REE)* and *Hyperbolic Error (HE)* for consistency of properties between original graph and coarsened graph. The details of these metrics are detailed in App. A.3.2. For DP, we also use membership inference attacks (Def. 3).

### 4.2 PRIVACY VS. ACCURACY TRADE-OFF

**Node-DP:** To compare our proposed framework with baselines for node-DP, we split the dataset into $80 - 20\%$ as training and test sets as given in (Zhang et al., 2024). With a given privacy budget $\varepsilon$ and fixed $\delta$ value chosen to be roughly equal to the inverse of each dataset's number of training nodes, we first obtain a coarsened graph via DPGC at 50% on which the GNN (GCN) model is trained.

Table 3 shows the test accuracies by benchmarked algorithms. DPGC demonstrates superior accuracy across all datasets for privacy budgets ranging from small to large ($\varepsilon = 1$ to 8). The results indicate that shifting the focus of privacy preservation to the coarsening process, rather than applying it directly during GNN training leads to enhanced performance. Furthermore, this approach allows DPGC to remain model and task agnostic, thereby offering greater flexibility in its application.
**Edge-DP:** Table 4 presents the results. The trend is consistent with the observations in Node-DP; DPGC outperforms all baselines across all datasets.

To further investigate how it behaves compared to baselines under different privacy budgets, we vary $\varepsilon = 1$ to 8 for node-DP and from 0.1 to 8 for edge-DP and compare the precision of the best-performing method. Figures 3 and 4 present the results for node-DP and edge-DP methods, respectively. For both (node and edge) privacy, DPGC consistently outperforms the best baselines from small to large privacy budgets. Furthermore, we observe that edge-DP has better accuracy than node-DP for the same values of $\varepsilon$. This is in line with the fact that nodes have more information to hide in a graph than edges.

### 4.3 ROBUSTNESS AGAINST MIA ATTACKS

This binary classification attack aims to infer if a node $v$ is in the training set $\mathcal{V}_T$ of the target GNN. Due to overfitting, GNNs often assign higher confidence scores to training nodes, which attackers exploit. The attacker first trains a shadow GNN on a dataset from the same distribution, with known membership labels, and then uses these scores to train a model that infers membership in the target graph. Following the TSTF approach (Olatunji et al., 2021),

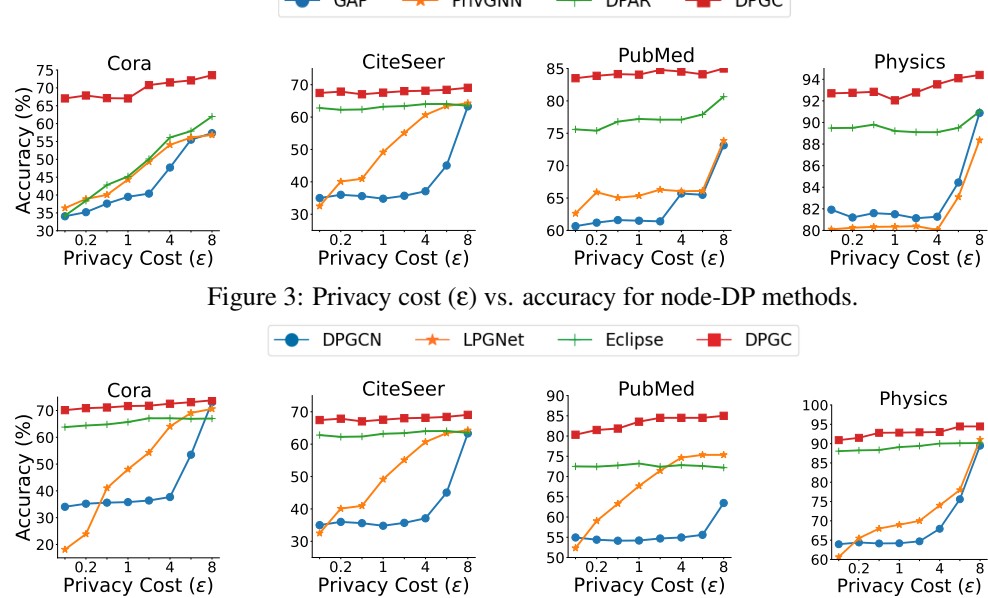

Figure 3: Privacy cost (ε) vs. accuracy for node-DP methods.

Figure 4: Privacy cost (ε) vs. accuracy for edge-DP methods.

we assume a strong adversary with a shadow dataset of 1,000 nodes per class, randomly sampled from the target dataset. The shadow model uses the same architecture as the target GNN. The attack model, a 3-layer MLP with 64 hidden units, is evaluated using AUC, averaged over 10 runs. Results are presented in Table 5; it is observed that the DPGC is as effective as other methods where attackers' accuracy is close to random guesses (approximately 50%).

### 4.4 ASSESSING THE QUALITY OF COARSENING

**Node Classification** Table 6 presents a comparison of DPGC for node classification when a GCN is trained on the coarsened graph (50%). DPGC outperforms baselines across all, but one dataset, by a significant margin. The superior performance of DPGC can be attributed to its use of WL-kernel, which differs

Table 5: Average AUC of node membership attack.

| Dataset | Method | $\varepsilon = 1$ | $\varepsilon = 2$ | $\varepsilon = 4$ | $\varepsilon = 8$ |
|---|---|---|---|---|---|
| Cora | GAP | $50.15 \pm 0.03$ | $50.20 \pm 0.05$ | $50.41 \pm 0.02$ | $51.08 \pm 0.05$ |
| | PrivGNN | $50.21 \pm 0.03$ | $50.21 \pm 0.03$ | $50.22 \pm 0.01$ | $50.19 \pm 0.03$ |
| | DPAR | $50.30 \pm 0.02$ | $50.64 \pm 0.02$ | $50.27 \pm 0.03$ | $50.62 \pm 0.03$ |
| | DPGC | $50.15 \pm 0.02$ | $50.20 \pm 0.02$ | $50.38 \pm 0.03$ | $50.58 \pm 0.05$ |
| CiteSeer | GAP | $50.04 \pm 0.10$ | $50.16 \pm 0.04$ | $50.33 \pm 0.04$ | $51.55 \pm 0.01$ |
| | PrivGNN | $50.03 \pm 0.01$ | $50.03 \pm 0.01$ | $50.60 \pm 0.05$ | $50.65 \pm 0.05$ |
| | DPAR | $50.25 \pm 0.03$ | $50.22 \pm 0.05$ | $51.31 \pm 0.03$ | $51.43 \pm 0.04$ |
| | DPGC | $50.05 \pm 0.02$ | $50.05 \pm 0.03$ | $50.26 \pm 0.03$ | $50.44 \pm 0.05$ |
| Pubmed | GAP | $50.06 \pm 0.04$ | $50.33 \pm 0.03$ | $51.10 \pm 0.02$ | $51.30 \pm 0.02$ |
| | PrivGNN | $50.05 \pm 0.03$ | $50.05 \pm 0.05$ | $50.33 \pm 0.01$ | $50.65 \pm 0.02$ |
| | DPAR | $50.10 \pm 0.02$ | $50.40 \pm 0.01$ | $50.45 \pm 0.05$ | $50.76 \pm 0.04$ |
| | DPGC | $50.31 \pm 0.02$ | $50.28 \pm 0.02$ | $50.53 \pm 0.08$ | $50.97 \pm 0.01$ |
| Coauthor | GAP | $50.05 \pm 0.05$ | $50.20 \pm 0.05$ | $50.50 \pm 0.05$ | $51.45 \pm 0.02$ |
| | PrivGNN | $50.01 \pm 0.03$ | $50.10 \pm 0.05$ | $50.55 \pm 0.03$ | $50.92 \pm 0.06$ |
| | DPAR | $50.30 \pm 0.05$ | $50.53 \pm 0.01$ | $50.97 \pm 0.03$ | $51.43 \pm 0.05$ |
| | DPGC | $50.25 \pm 0.02$ | $50.33 \pm 0.09$ | $50.30 \pm 0.05$ | $50.41 \pm 0.05$ |

from the connectivity-based approaches commonly employed in other methods. Message-passing GNNs capture locality in their embeddings, meaning that two nodes that are distant in the graph may still have similar representations. Connectivity-based coarsening methods overlook this aspect, while DPGC, by leveraging the WL-kernel, aligns well with the message-passing structure of GNNs and effectively captures these subtle relationships.

**Robustness to other GNN architectures:** In Table 7, we benchmark the performance of DPGC on four different GNN architectures, namely GCN (Kipf & Welling, 2016a), GAT (Veličković et al., 2017), GIN (Xu et al., 2018), GRAPHSAGE Hamilton et al. (2017). We observe that across all datasets and architectures, DPGC produces either the highest or second highest accuracy. Our results demonstrate the robustness of using Weisfeiler-Lehman (WL) kernel embeddings as the signal for supernode construction, in contrast to the dominant strategy in the literature that relies on network connectivity. From Def. 5 and Def. 6, we know that regardless of the GNN architecture, similar $L$-hop ego neighborhoods lead to similar embeddings. We exploit these observations by employing the WL-kernel to construct embeddings that characterize the L-hop neighborhood of a node. This method allows us to group nodes into supernodes based on the similarity of their extended local structures,

Table 6: Node classification Accuracy (ACC) when a GCN is trained on 50% coarsened graph (higher ACC is better, with boldface indicating the best results).

| Model | Cora | CiteSeer | PubMed | Physics | Coauthor | DBLP |
|-------|------|----------|--------|---------|----------|------|
| **LVN** | $79.70 \pm 1.20$ | $69.54 \pm 2.33$ | $77.87 \pm 1.22$ | $93.74 \pm 0.5$ | $85.90 \pm 0.22$ | $77.05 \pm 0.30$ |
| **LVE** | $81.57 \pm 0.92$ | $70.60 \pm 1.12$ | $78.34 \pm 0.78$ | $93.86 \pm 0.54$ | $\mathbf{87.63 \pm 0.71}$ | $78.72 \pm 1.02$ |
| **LVC** | $80.92 \pm 0.45$ | $68.81 \pm 1.42$ | $73.32 \pm 3.40$ | $92.94 \pm 0.62$ | $85.66 \pm 0.21$ | $78.69 \pm 0.28$ |
| **HEM** | $79.90 \pm 1.51$ | $71.11 \pm 1.34$ | $74.66 \pm 2.12$ | $93.03 \pm 0.47$ | $69.54 \pm 3.22$ | $77.46 \pm 1.23$ |
| **Alg. Distance** | $79.83 \pm 1.05$ | $70.09 \pm 0.73$ | $74.59 \pm 1.05$ | $93.94 \pm 0.09$ | $83.74 \pm 1.03$ | $74.51 \pm 0.30$ |
| **Affinity** | $80.20 \pm 2.31$ | $70.70 \pm 1.16$ | $80.53 \pm 0.73$ | $93.06 \pm 0.44$ | $85.10 \pm 0.25$ | $78.15 \pm 0.19$ |
| **Kron** | $80.71 \pm 1.76$ | $69.00 \pm 2.15$ | $74.89 \pm 2.91$ | $92.26 \pm 0.82$ | $84.22 \pm 0.04$ | $77.79 \pm 0.08$ |
| **FACH** | $74.92 \pm 0.13$ | $66.97 \pm 0.20$ | $85.65 \pm 0.07$ | $94.70 \pm 0.05$ | $74.19 \pm 1.51$ | $75.50 \pm 0.33$ |
| **DPGC** | $\mathbf{85.33 \pm 0.10}$ | $\mathbf{73.25 \pm 0.16}$ | $\mathbf{85.91 \pm 0.51}$ | $\mathbf{95.85 \pm 0.11}$ | $86.03 \pm 0.14$ | $\mathbf{78.75 \pm 0.23}$ |

Table 7: Node classification Accuracy (ACC) different GNN model (GCN, GAT, GIN and GraphSage) when trained on 50% coarsened dataset (higher ACC is better). The best and the second best results in row are highlighted in bold font and underlining, respectively

| Dataset | GNN | LVN | LVE | LVC | HEM | Alg. Distance | Affinity | Korn | FACH | DPGC |
|---------|-----|-----|-----|-----|-----|---------------|----------|------|------|------|
| **Cora** | **GCN** | 79.70 | 81.57 | 80.92 | 79.90 | 79.83 | 80.20 | 80.71 | 74.92 | **85.33** |
| | **GAT** | 69.50 | 74.02 | **74.52** | 68.85 | 73.10 | 73.63 | 73.24 | 73.21 | 74.21 |
| | **GIN** | 47.75 | 35.69 | 53.10 | 35.45 | 63.20 | 25.40 | 48.90 | 65.25 | **66.13** |
| | **GraphSage** | 70.49 | 69.42 | 70.11 | 69.20 | 71.96 | 67.80 | **73.25** | 68.35 | 70.50 |
| **PubMed** | **GCN** | 77.87 | 78.34 | 75.32 | 74.66 | 74.59 | 80.53 | 74.89 | 85.65 | **85.91** |
| | **GAT** | 75.20 | 72.54 | 74.76 | 61.05 | 70.35 | 60.70 | 71.95 | 81.20 | **83.54** |
| | **GIN** | 74.70 | 40.28 | 47.20 | 36.05 | 33.15 | 49.75 | 40.45 | 74.50 | **75.35** |
| | **GraphSage** | 78.75 | 63.55 | 67.20 | 60.21 | 64.05 | 71.20 | 63.25 | 80.50 | **83.66** |
| **DBLP** | **GCN** | 77.05 | 78.42 | 78.69 | 77.46 | 74.51 | 78.15 | 77.79 | 75.50 | **78.75** |
| | **GAT** | 70.20 | **74.00** | 72.80 | 71.35 | 71.15 | 71.12 | 72.25 | 73.49 | 73.95 |
| | **GIN** | 35.85 | 33.96 | 35.24 | 25.16 | 51.47 | 47.30 | 42.25 | 53.55 | **55.29** |
| | **GraphSage** | 68.55 | 60.22 | **73.31** | 72.70 | 72.18 | 71.79 | 71.75 | 73.22 | 73.22 |
| **Physics** | **GCN** | 93.74 | 93.86 | 92.94 | 93.03 | 93.94 | 93.06 | 92.26 | 94.70 | **95.85** |
| | **GAT** | 91.05 | 91.70 | 91. 44 | 91.40 | 91.94 | 92.30 | 91.55 | 92.20 | **93.40** |
| | **GIN** | 90.30 | 87.90 | 89.35 | 90.12 | 87.56 | 91.22 | 91.55 | 91.20 | **92.50** |
| | **GraphSage** | 89.90 | 87.55 | 87. 33 | 89.91 | 87.55 | 90.12 | 91.42 | **93.35** | 93.22 |

rather than just immediate connections. The effectiveness of this approach is evidenced by its performance across various GNN prediction tasks, showing that capturing higher-order neighborhood similarities can lead to more effective graph coarsening for GNN applications.

**Additional Experiments:** In Fig. 7 in the Appendix, we present further evidence that even after 70% coarsening for training, DPGC maintains its accuracy. Furthermore, we show that DPGC continues to outperform existing coarsening algorithms in the task of Link Prediction (Table 9, App. A.4.1). We also evaluate the relative eigen error of DPGC in App. A.4.2 Table 10. In the App A.4.3, we further investigate the relationship between graph size reduction and privacy budget ε.

## 5 CONCLUSIONS, LIMITATIONS AND FUTURE WORKS

Graph Neural Networks (GNNs) face two major obstacles to widespread adoption: potential exposure of sensitive training data in public applications and computational challenges due to vast datasets with millions of nodes and edges. Our algorithm, Differentially Private Coarse Graining (DPGC), tackles both issues simultaneously through privacy-preserving graph compression. DPGC consistently outperformed existing methods in both graph coarsening and differential privacy across six real-world datasets. This superior performance stems from our novel approach of using the Weisfeiler-Lehman (WL) kernel for supernode construction, rather than conventional network connectivity metrics. This strategy enables provable privacy guarantees while enhancing computational efficiency. By addressing privacy concerns and reducing computational demands, DPGC paves the way for broader, safer deployment of GNNs in real-world applications.

**Limitations and Future Works:** In this work, we base the methodology on message-passing GNNs. It remains to be seen how it generalizes to other forms of GNNs such as Graph Transformers.

## Reproducibility

To replicate the results reported in this paper, the software and hardware requirements are given in the Appendix A.3.1. All the data sets used in the experiments are publicly available in the PyTorch Geometric Library. Parameters, hyperparameters and instructions for running code are also given in the codebase at `https://anonymous.4open.science/r/DPGC-6BE8`.

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

# A  APPENDIX

## A.1  EXPLANATION OF LOCALITY-AWARENESS IN FIG. 1

GNNs iteratively accumulate messages (features) from their neighbors in a layer-by-layer manner (Recall Section 3.1). A computation tree, also known as the receptive field, encodes how messages propagate to the target node, whose embedding is being computed. In the context of $v_1$ and $v_{11}$ in Fig. 1, the 2-hop computation trees are isomorphic for $v_1$ and $v_1 1$ leading to identical embeddings in any 2-layered message-passing GNN regardless of the specific architecture being used. The computation trees are identical, since their one-hop and two-hop nodes have identical attributes.

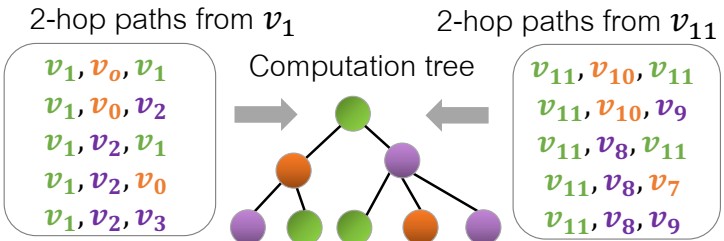

Figure 5: Computation trees of $v_1$ and $v_{11}$ in Fig. 1.

In our proposed algorithm, through the usage of the WL kernel (Eq. 3), we emulate this message-passing framework and thereby encode locality information into the coarsening process. In other words, if two nodes have similar computation trees, they are likely to be coarsened into a supernode. This locality-aware coarsening represents a significant departure from existing coarsening methods in the literature. By considering the similarity of local structures rather than just graph proximity, our approach can identify and merge nodes that play similar roles in the graph, even if they are distant. This novel method enables us to achieve superior results compared to traditional coarsening techniques.

## A.2  TIME COMPLEXITY AND RUN TIME

**Complexity Analysis**: The DPGC algorithm consists of the following phases. We assume $n$ and $m$ to be the number of nodes and edges, respectively, in the original graph.

1. **Embedding ego graphs:** The first phase involves computing node embeddings using WL-kernel. WL-kernel performs the same iterative message passing as in any message-passing GNNs, where each edge is traversed twice. Hence, the time complexity is $O(m)$
2. Projection into buckets using LSH: The complexity of performing projections is $O(nJd)$ time, where $d$ is the feature dimension, and $J$ is the number of hash functions. This complexity arises from the multiplication of matrices of size $n \times d$ and $d \times J$.
3. **Construction of supernodes and superedges:** Constructing supernodes consumes $O(nJ)$ time since it involves iterating over all nodes and identifying the most frequent bucket, among $J$ buckets, they have been hashed into. Once supernodes are constructed, we need to iterate over all original edges and see if the two endpoints are in two different supernodes, in which case an edge is added among the corresponding supernodes. This requires $O(m)$ time.
4. **Learning node attributes:** Finally, we learn the node attributes, which involves minimizing Eq. 5. requiring $O(nkd)$ time. Since the Laplacian is fixed for the coarsened graph and the multiplication of $\tilde{X}$ by $C^T LC$ takes $O(k^2 d)$. The trace operation has complexity $O(kd)$. For the second term, $C\tilde{X}$ takes $O(nkd)$ time and computing the Frobenius norm involves subtraction and squaring, which takes $O(nd)$ time. Therefore, the overall time complexity for node feature learning is dominated by the term $O(nkd)$ as $k$ is small.

After combining all components, the overall time complexity is $O(nJd + m)$. The runtime comparison of coarsening methods is given in Table 8.

Table 8: Runtime (in seconds) comparison of graph coarsening methods. The results are presented for coarsening the graph to 50% of its original size, averaged over 5 runs.

| Model | Cora | CiteSeer | PubMed | Physics | Coauthor | DBLP |
|---|---|---|---|---|---|---|
| **LVN** | 6.63 | 8.70 | 25.00 | 58.50 | 23.41 | 22.80 |
| **LVE** | 5.35 | 7.40 | 18.79 | 68.20 | 16.81 | 20.61 |
| **LVC** | 7.30 | 9.80 | 61.85 | 70.00 | 24.56 | 38.40 |
| **HEM** | 0.70 | 1.45 | 12.05 | 40.50 | 7.64 | 8.42 |
| **Alg. Distance** | 0.95 | 1.55 | 10.50 | 46.55 | 9.59 | 9.75 |
| **Affinity** | 2.40 | 2.55 | 165.40 | 910.50 | 165.10 | 115.5 |
| **Kron** | 0.65 | 1.40 | 6.00 | 35.65 | 9.01 | 7.10 |
| **FACH** | 0.53 | 0.75 | 1.65 | 6.05 | 3.20 | 1.50 |
| **DPGC** | 0.53 | 0.76 | 1.67 | 6.10 | 3.15 | 1.65 |

## A.3 QUALITY OF GRAPH COARSENING: NODE CLASSIFICATION AND LINK PREDICTION ALGORITHMS

### A.3.1 EXPERIMENTAL SETUP

All experiments were performed on a machine with an Intel(R) Core(TM) CPU @ 2.30GHz, 16GB RAM, and an RTX A4000 GPU with 16GB memory, running Microsoft Windows 11 HSL. To ensure consistency with all baselines, we used two hidden-layer GNN models with standard hyperparameter values, following Kipf & Welling (2016b); Huang et al. (2021). We used a train-validation-test split of 80:10:10 in our experiments. The dimensions of both hidden layer embeddings were set to 16, with a learning rate of 0.005 and a weight decay rate of $5 \times 10^{-4}$.

### A.3.2 REE AND HE

A popular metric to evaluate the quality of the coarsened graph is **Relative Eigen Error (REE)** Loukas (2019). The coarsened graphs are the best approximation of the original if the value of REE is close to zero. Similarly, to quantify how much structural similarities are preserved in the coarsened graph $\mathcal{G}_c$, **Hyperbolic error (HE)** is defined in terms of lifted Laplacian $L_l$.

**Definition 10** (Lifted Laplacian $\mathbf{L}_f$ Loukas & Vandergheynst (2018)). *For the coarsening matrix $\mathbf{P} \in \mathbb{R}_+^{k \times n}$ and the Laplacian matrix for coarsened graph $\tilde{\mathbf{L}} \in \mathbb{R}^{k \times k}$, the lifted Laplacian matrix is defined as follows:*

$$\mathbf{L}_f = \mathbf{P}^T \tilde{\mathbf{L}} \mathbf{P}.$$

The lifted Laplacian matrix reconstructs the original dimension $n \times n$ from the coarsened dimension of $k \times k$. When moving from a $\tilde{\mathcal{G}}$ representation back to the $\mathcal{G}$, the disparity between the original and projected data can be quantified by the Hyperbolic Error (HE).

**Definition 11** (Hyperbolic Error (HE) Loukas (2019)). *For the given feature matrix $\mathbf{X}$, the hyperbolic error between the original Laplacian matrix $\mathbf{L}$ and lifted Laplacian matrix $\mathbf{L}_f$ is defined as*

$$HE = \text{arccosh}\left(1 + \frac{||(\mathbf{L} - \mathbf{L}_f)\mathbf{X}||_F^2 ||\mathbf{X}||_F^2}{2\text{Tr}(\mathbf{X}^T\mathbf{L}\mathbf{X}) \cdot \text{Tr}(\mathbf{X}^T\mathbf{L}_f\mathbf{X})}\right).$$

Along with these metrics, one can further perform downstream tasks on a coarsened graph and compare it with the original one, e.g. node classification using GNNs. In Table 10, we have presented REE for all algorithms at 50% coarsening ratio. In Figure 6, REE and HE values have been evaluated on different coarsening ratios ranging from 10% to 95% on the PubMed dataset. The DPGC is comparable with the state-of-the-art coarsening algorithm on REE and HE while having better downstream take performance and privacy protection. Table 6 compares the node classification accuracy of GCN when graph size is reduced to 50% using coarsening algorithms. To further show the effectiveness of DPGC, we coarsened the graph from 5% to 95% using all baselines and then trained GCN, Figure 6 (right) shows that DPGC consistently performs better in comparison to the state-of-the-art method.

As DPGC is designed for graph coarsening, we can use any GNN architecture to perform downstream tasks. To establish this, we performed node classification using GAT Veličković et al. (2017), GIN Xu

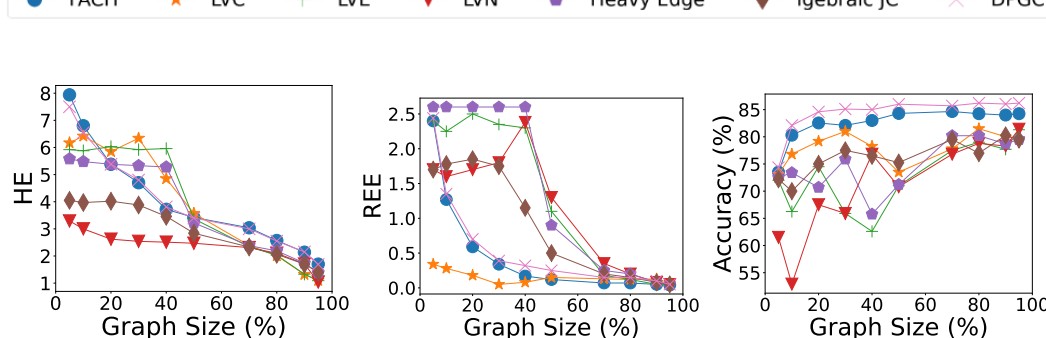

Figure 6: Coarsening quality: HE (left), REE (middle) and GCN node classification accuracy (right) comparison of coarsening algorithms on PubMed dataset.

Table 9: Link prediction performance of the baselines and DPGC in terms of AUC (higher is better). The coarsening ratio is kept small for relatively small datasets (Cora & CiteSeer) and large for larger datasets (PubMed). Here, size denotes the size of the coarsened graph. The boldface indicates the best results.

| Dataset | Size | Full Data | GCond | LVN | FGC | LAGC | DPGC |
|---|---|---|---|---|---|---|---|
| Cora | 30% | 84.14 ±0.77 | 68.13 ± 0.16 | 70.42 ± 0.55 | 77.35 ± 0.22 | 78.27 ± 0.28 | **82.25 ± 0.05** |
| | 10% | | 66.24 ± 0.54 | 68.06 ± 0.24 | 75.27 ± 0.15 | 77.35 ± 0.37 | **80.03 ± 0.14** |
| | 5% | | 63.46 ± 1.09 | 63.19 ± 0.97 | 73.12 ± 0.22 | 75.28 ± 0.14 | **78.00 ± 0.02** |
| CiteSeer | 30% | 78.46 ± 0.60 | 72.18 ± 0.48 | 71.70 ± 0.23 | 73.16 ± 0.16 | 75.25 ± 0.19 | **77.00 ± 0.02** |
| | 10% | | 69.82 ± 0.40 | 69.68 ± 0.59 | 70.01 ± 0.37 | 74.02 ± 0.33 | **75.50 ± 0.01** |
| | 5% | | 63.11 ± 0.80 | 64.12 ± 0.66 | 68.01 ± 0.37 | 72.29 ± 0.44 | **72.46 ± 0.05** |
| PubMed | 5% | 83.46 ± 0.59 | 61.51 ± 0.85 | 62.32± 0.87 | 67.27 ±0.38 | 77.25 ± 0.31 | **81.05 ± 0.05** |
| | 3% | | 60. 81 ± 0.60 | 62.36 ± 0.58 | 66.50 ± 0.20 | 72.48 ± 0.22 | **76.00 ± 0.05** |
| | 1% | | 57.14 ± 0.55 | 61.78 ± 0.22 | 66.05 ± 0.40 | 68.34 ± 0.46 | **72.40 ± 0.01** |

et al. (2018), and GraphSage Hamilton et al. (2017)) popular GNN architectures on four datasets when the graph size is reduced to 50%. Table 7 presents the results on different datasets for four GNN architectures. The performance of all GNNs is observed to be similar as it is on original datasets.

## A.4 ADDITIONAL EXPERIMENTS

### A.4.1 LINK PREDICTION

In link prediction, the task is to predict the existence of a connection between two given nodes. We use SEAL (Zhang & Chen, 2018) on the coarsened graph for this task. Table 9 presents the AUCROC achieved by DPGC and the baselines at various coarsening levels. DPGC outperforms all baselines across all coarsening ratios demonstrating its robustness and efficacy.

### A.4.2 RELATIVE EIGEN ERROR (REE)

A common strategy among baselines has been to create a coarsened graph with a similar eigen-spectrum as the original graph. Consequently, REE has become a popular metric for assessing coarsening quality (See App. A.3.2 for the mathematical formulation).

Table 10 presents a comparison of REE values for the top-100 eigenvalues across all baseline methods. Before analyzing the results, we note that while a low REE is likely to correlate to good GNN training on the coarsened graph, this is not a necessity. As previously discussed, in a message-passing GNN, a node's embedding encapsulates information from its L-hop ego graph. This means that nodes do not necessarily need

Table 10: The Relative Eigen Error (REE) of DPGC and baselines at 50% coarsening ratio. (Lower is better).

| Model | Cora | Citeseer | PubMed | Physics | Coauthor | DBLP |
|---|---|---|---|---|---|---|
| LVN | 0.12 | 0.18 | 0.11 | 0.27 | 0.25 | 0.12 |
| LVE | 0.13 | 0.14 | 0.96 | 0.04 | 0.05 | 0.14 |
| LVC | 0.09 | 0.06 | 1.21 | 0.04 | **0.03** | 0.08 |
| HEM | 0.07 | 0.04 | 0.83 | 0.03 | 0.05 | 0.09 |
| Alg. Dist | 0.11 | 0.11 | 0.40 | 0.12 | 0.09 | **0.05** |
| Affinity | 0.10 | 0.06 | **0.06** | 0.05 | 0.06 | 0.07 |
| Kron | **0.07** | **0.03** | 0.38 | 0.06 | 0.06 | 0.06 |
| FACH | 0.22 | 0.34 | 0.18 | 0.02 | 0.21 | 0.15 |
| DPGC | 0.13 | 0.23 | 0.16 | **0.02** | 0.18 | 0.10 |

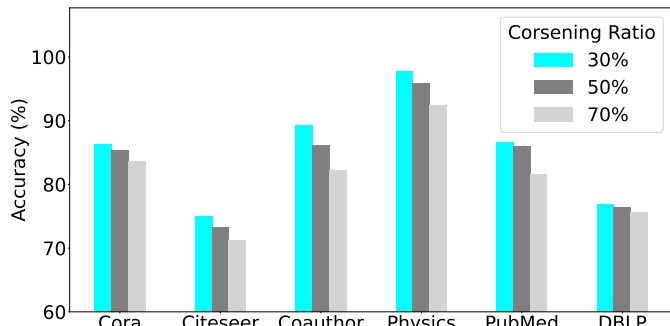

Figure 7: Effect of the coarsening process on GNN prediction performance when coarsening all graph datasets with different coarsening ratios of 30%, 50% and 70% in DPGC.

to be spatially close in the network to have similar embeddings. Coarsening strategies that focus on preserving the eigen-spectrum tend to group only proximal nodes into supernodes, which may not always be optimal for preserving the most relevant graph properties for GNN training.

The above intuition aligns well with the results presented in Table 10. Although the baselines exhibit lower RRE than DPGC in most datasets, this does not translate to superior performance in GNN prediction tasks (Table 6 and Table 9).

### A.4.3 PRIVACY AND REDUCTION IN GRAPH SIZE

In this section, we further investigate the relationship between graph size reduction using DPGC and a given privacy budget $\varepsilon$, while ensuring utility remains preserved. In Fig. 8, we observe that for all datasets, the reduction in graph size is proportional to the privacy budget $\varepsilon$. Specifically, when $\varepsilon$ is small, the graph size should be reduced less, while for larger $\varepsilon$, the graph can be coarsened to a greater extent without significant loss of utility in downstream tasks such as node classification.

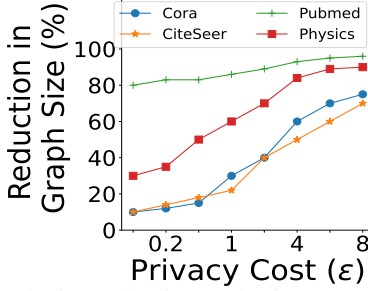

Figure 8: Trade-off between graph size reduction and privacy cost while maintaining downstream utility at the same level.

