# OpenReview forum: "Task and Model Agnostic Differentially Private Graph Neural Networks via Coarsening"
_ICLR.cc/2025/Conference — Submitted to ICLR 2025_

### Official Review · Reviewer_RXHP · 2024-10-25

**Soundness:** 1
**Presentation:** 3
**Contribution:** 3
**Rating:** 1
**Confidence:** 5

**Summary:**

The submission proposes a differentially private procedure for releasing coarsened graphs, which can then be used for downstream tasks like graph neural network training.

The coarsening procedure, which maps a graph to a smaller graph in which each node corresponds to a set of nodes in the original graph, involves multiple steps. First, node attributes are embedded via a diffusion step with skip connections ("WL-Kernel"). Then, these embeddings are clustered via locality sensitive hashing (LSH), with each cluster corresponding to a node in the coarsened graph. A new adjacency is defined in the usual manner for edge contractions, i.e., two clusters are connected if any of their components are connected. Finally, a new attribute matrix is determined via gradient-based optimization of an objective that enforces similar energy to the original graph, i.e., edge-weighted squared differences between attributes.

To achieve edge- or node-level differential privacy, calibrated Gaussian noise is added (1) after the LSH projection function and (2) to the final attribute matrix.

Finally, the proposed method is evaluated by comparing its privacy-utility tradeoff to differentially private GNN architectures (e.g. GAP) and graph-specific variations of DP-SGD (e.g. PrivGNN) in an inductive node classification setting. In addition, the effectiveness of a membership inference attack on these approaches is tested and the proposed coarsening procedure is benchmarked against prior work on graph coarsening.

**Strengths:**

* The concept of releasing an entire graph with formal node-level privacy guarantees, rather than trying to develop GNN-specific procedures, is exciting and well-motivated
* The individual components are justified via references to earlier theoretical works on, e.g., GNN expressivity or graph spectral similarity
* Even ignoring the differential privacy aspect, the coarsening procedure appears to represent an improvement over prior work, see Tables 1&6 (I am not sufficiently familiar with coarsening literature to make a definite statement on this).
* The experimental evaluation rigorously explores the space of DP-GNN baselines, coarsening methods, and downstream GNNs.
* Results are reported with standard deviations.

**Weaknesses:**

### Primary weakness
The main weakness of the proposed work is that **the privacy analysis is incorrect on various levels**. Specifically, (1) the privacy guarantees for the proposed procedure's individual components are overly optimistic or incorrect, (2) privacy leakage of the composition of these components is underestimated and (3) **the overall procedure does in fact not provide any form of edge- or node-level privacy** (contrary to claims in the paper).

* The privacy analysis of the noisy LSH step reuses a result from [1], whose sensitivity analysis assumes that a single row in the input matrix changes. However, due to the WL-kernel diffusion, changes to a single row in the attribute matrix can change all rows in the embedding matrix that is projected by LSH (e.g., in a complete graph). The added noise is thus too small. A valid analysis would require using the group privacy property with group size $N$.
* The privacy analysis for the final attribute matrix relies on the claim that the involved optimization problem of the form $\min_\tilde{X} f(\alpha, X, \tilde{X})$ had sensitivity $\alpha$. It is not clear why the solution of this non-linear minimization problem should have sensitivity $\alpha$.
* The proposed DPGC procedure does not actually solve this optimization problem in closed form (which the privacy analysis of the manuscript assumes), but uses gradient descent. Since the considered neighboring relation does not constrain how much attribute matrix $X$ can change,  each gradient can change arbitrarily, i.e., the global sensitivity is $\infty$.
* In addition, the analysis ignores that each gradient step accesses the private attribute matrix, meaning the privacy guarantees should weaken with the number of steps. A valid analysis would require use of differentially private stochastic gradient descent, alongside composition or amplification-by-iteration analysis.
* Assuming the previous analysis were correct, the LSH and the attribute learning step would each be $(\epsilon,\delta)$. This does not imply that the sequential composition of these steps is $(\epsilon,\delta)$ DP. One needs to apply some composition theorem, e.g., [2].
* **The adjacency of the coarsened graphs is given by a contraction of the original adjacency. No steps are undertaken to prevent leakage of the adjacency matrix through the contraction operation.** For instance, it is trivial for an adversary to distinguish a graph with $0$ edges and a graph with $1$ edge (both of which are considered neighboring in edge- and node-level DP).

### Other weaknesses
* The discussion of prior work on differentially private GNNs focuses on methods that attempt to learn private embeddings. It omits works on edge-level DP that, similar to this work, focus on making the input graph/edges themselves private, e.g., [4] or LapGraph from [5]. It also omits works that use DP versions of personalized pagerank to construct a privacy-preserving adjacency matrix, e.g., [6], [7].
* The manuscript does not discuss how to determine labels for the coarsened graph and does not propose a procedure for ensuring privacy of the nodes' labels.
* Parts of the main Figure 2 are not representative of the proposed method. Specifically, the "lock" symbol above "5. Supernode's edge assignment" suggests that there was some procedure that protected adjacency information, which is not the case.

### Minor comments
* It would be nice to provide a definition of node-level privacy (like the one for edge-level privacy in ll.144-146). Some works only assume that the number of nodes is constant and only the features and edges of a single node change arbitrarily, while other works assume that nodes can be entirely removed.
* The method appears to be limited to the inductive setting, where we have a separate training graph that is coarsened for DP training. It is unclear how this method can be applied to the more common transductive setting. That is, how we can provide predictions for a partially labelled original graph $G$ after training a model on a coarsened version $\tilde{G}$ of $G$?
* The results for GAP are identical in the node-DP (Table 3) and edge-DP (Table 4) setting. However, since its privacy guarantees are stronger for edge-DP, the accuracies in the edge-DP setting at any given privacy budget should be higher.
* The GAP baseline has been superseded by ProGAP [3], which enables multiple message passing steps. One may want to include it as a baseline (I do not expect the authors to do this for the rebuttal).
* The chosen delta ($2 \times 10^{-3}$) is quite large, considering that the considered datasets have over $10^3$ nodes. Usually, one would choose $\delta \ll 1 \mathbin{/}N$.

---

[1] Kenthapadi et al. Privacy via the Johnson-Lindenstrauss transform. Journal of Privacy and Confidentiality. 2013.
[2] Kaiorouz et al. The Composition Theorem for Differential Privacy. ICML 2014.
[3] Sajadmanesh et al. ProGAP: Progressive Graph Neural Networks with Differential Privacy Guarantees. WSDM 2024.
[4] Vu et al. Privacy-Preserving Visual Content Tagging using Graph Transformer Networks. MM 2020.
[5] Wu et al. LINKTELLER: Recovering Private Edges from Graph Neural Networks via Influence Analysis. 2022 IEEE Symposium on Security and Privacy (SP).
[6] Epasto et al. Differentially Private Graph Learning via Sensitivity-Bounded Personalized PageRank. NeurIPS 2022.
[7] Wei at al. Differentially Private Graph Diffusion with Applications in Personalized PageRanks. NeurIPS 2024.

---

Given the listed weaknesses, specifically the lack of privacy protection, **I recommend rejection**.
I nevertheless think that the underlying coarsening method could be a meaningful contribution to the field of graph machine learning.
I would encourage the authors to either focus exclusively on coarsening without DP, or to try and eliminate the remaining sources of privacy leakage before resubmitting to another venue.

**Questions:**

* How are labels for the coarsened graph determined in your experiments?

---

### Official Review · Reviewer_V4vp · 2024-10-26

**Soundness:** 3
**Presentation:** 3
**Contribution:** 2
**Rating:** 6
**Confidence:** 3

**Summary:**

The author used the graph coarsening technique to help to deal with the high computational costs during training or struggle to generalize across various GNN models and task objectives. The method achieves superior compression accuracy trade-offs while maintaining robust privacy guarantees, outperforming state-of-the-art baselines in this domain.

**Strengths:**

1. The idea is very interesting.
2. The privacy guarantee on DP technique is very solid.

**Weaknesses:**

1. The two challenges that proposed to solve by the author: scalability and privacy guarantees are not related with each other.
2. DP is meant to deal with the privacy attack. However, this does not mentioned in the paper. Attacks including the poisoning attacks, and this cannot be solved by the technique in the paper.

**Questions:**

See the weaknesses listed above.

---

### Official Review · Reviewer_zxEg · 2024-11-03

**Soundness:** 2
**Presentation:** 3
**Contribution:** 2
**Rating:** 5
**Confidence:** 4

**Summary:**

The paper proposes Differentially Private Graph Coarsening (DPGC), a method with strong generalizability that can be applied to all downstream GNNs.

**Strengths:**

The paper is well-written and easy to follow, with a logical flow that enhances comprehension. The experiment details are re clearly demonstrated and the code is released, ensuring that the methodology is transparent and reproducible, which further reinforces the reliability of the study.

**Weaknesses:**

The paper claims that the proposed private GNN meets node-DP; however, node-level DP in GNNs requires the protection of node features, all links, and node labels. It appears that the proposed method does not provide protection for node labels. This results in two major concerns:

1. The claim of satisfying node-level DP is not valid, as label protection is essential to uphold this standard.
2. The experiments presented in Table 3 are potentially unfair. Methods such as DP-MLP, DP-GNN, GAP, PrivGNN, and DPAR include protection for node labels, which raises concerns about the comparability of the results. The observed performance improvement might be attributed to the lack of label protection, making it unclear whether the reported gains are due to the omission of label protection.

**Questions:**

Please see weakness. If my concerns are resolved, I am willing to raise the score.

---

### Comment · Area_Chair_HoJc · 2024-11-28

I would like to encourage the reviewers to engage with the author's replies if they have not already done so. At the very least, please
acknowledge that you have read the rebuttal.

---

### Meta-Review · Area_Chair_HoJc · 2024-12-19

**Metareview:**

The authors propose a method for releasing coarsened graphs with DP guarantees, combining a diffusion step (WL-kernel) with noisy LSH-based clustering. Reviewer RXHP claims that the privacy analysis is incorrect on various levels and that "the overall procedure does in fact not provide any form of edge- or node-level privacy". I agree with this assessment. Reviewer zxEg also states that the "claim of satisfying node-level DP is not valid".

**Additional Comments On Reviewer Discussion:**

The authors did not make an attempt to address the reviewers concerns (did not reply.

---

### Decision · Program_Chairs · 2025-01-22

Reject